# Harnessing *Mycobacterium bovis* BCG Trained Immunity to Control Human and Bovine Babesiosis

**DOI:** 10.3390/vaccines10010123

**Published:** 2022-01-14

**Authors:** Reginaldo G. Bastos, Heba F. Alzan, Vignesh A. Rathinasamy, Brian M. Cooke, Odir A. Dellagostin, Raúl G. Barletta, Carlos E. Suarez

**Affiliations:** 1Department of Veterinary Microbiology and Pathology, College of Veterinary Medicine, Washington State University, Pullman, WA 99164-7040, USA; heba.alzan@wsu.edu; 2Parasitology and Animal Diseases Department, Veterinary Research Institute, National Research Center, Giza 12622, Egypt; 3Australian Institute of Tropical Health and Medicine, James Cook University, Cairns, QLD 4870, Australia; vignesh.ambothirathinasamy@jcu.edu.au (V.A.R.); brian.cooke@jcu.edu.au (B.M.C.); 4Centro de Desenvolvimento Tecnológico, Universidade Federal de Pelotas, Pelotas 96010-900, Rio Grande Do Sul, Brazil; odir@ufpel.edu.br; 5School of Veterinary Medicine and Biomedical Sciences, University of Nebraska-Lincoln, Lincoln, NE 68583-0905, USA; rbarletta@unl.edu; 6Animal Disease Research Unit, United States Department of Agriculture-Agricultural Research Service, Pullman, WA 99164-7040, USA

**Keywords:** *Mycobacterium bovis* bacillus Calmette-Guerin (BCG), trained immunity, recombinant BCG, *Babesia* spp., *Babesia bovis*, *Babesia microti*, human babesiosis, bovine babesiosis, anti-*Babesia* vaccine

## Abstract

Babesiosis is a disease caused by tickborne hemoprotozoan apicomplexan parasites of the genus *Babesia* that negatively impacts public health and food security worldwide. Development of effective and sustainable vaccines against babesiosis is currently hindered in part by the absence of definitive host correlates of protection. Despite that, studies in *Babesia microti* and *Babesia bovis*, major causative agents of human and bovine babesiosis, respectively, suggest that early activation of innate immune responses is crucial for vertebrates to survive acute infection. Trained immunity (TI) is defined as the development of memory in vertebrate innate immune cells, allowing more efficient responses to subsequent specific and non-specific challenges. Considering that *Mycobacterium bovis* bacillus Calmette-Guerin (BCG), a widely used anti-tuberculosis attenuated vaccine, induces strong TI pro-inflammatory responses, we hypothesize that BCG TI may protect vertebrates against acute babesiosis. This premise is supported by early investigations demonstrating that BCG inoculation protects mice against experimental *B. microti* infection and recent observations that BCG vaccination decreases the severity of malaria in children infected with *Plasmodium falciparum*, a *Babesia*-related parasite. We also discuss the potential use of TI in conjunction with recombinant BCG vaccines expressing *Babesia* immunogens. In conclusion, by concentrating on human and bovine babesiosis, herein we intend to raise awareness of BCG TI as a strategy to efficiently control *Babesia* infection.

## 1. Introduction

Babesiosis is a tick-borne disease of vertebrates caused by apicomplexan hemoprotozoan parasites of the genus *Babesia* [1,2]. *Babesia* parasites have a complex lifecycle that involves the development of asexual stages in vertebrate hosts and sexual stages in tick vectors [1,3,4]. In vertebrate hosts, *Babesia* has evolved to invade and replicate exclusively inside red blood cells (RBCs) [1]. Since RBCs lack the machinery to process and present antigens, this strategy provides a relatively safe environment for the parasite to evade the vertebrate immune system. Considered an emerging disease in humans and an economically important condition for cattle, human and bovine babesiosis can adversely affect public health and food security, respectively [1,2,3,4,5,6,7,8]. Babesiosis in humans is predominantly caused by *Babesia microti* in the USA and *Babesia divergens* in Europe [9,10]. Clinical signs of human babesiosis vary from mild flu-like symptoms to severe disease, especially in immunocompromised individuals, which are characterized by renal failure, acute respiratory distress, and disseminated intravascular coagulations [11]. No vaccines are currently available to protect humans against babesiosis. Moreover, the emerging situation of the disease in the USA is worsened by the recent appearance of competent ticks for *B. microti* in previously tick-free areas [12,13,14]. The use of anti-babesial drugs, such as azithromycin and atovaquone, is the only current option to treat human babesiosis. However, the development of resistance to these therapeutics and the potential transmission of parasites, especially *B. microti*, via blood supply are major concerns [9,15,16].

Several *Babesia* species infect and cause disease in domestic and wild animals [8,17]. In this review, we focus on bovine babesiosis caused by *B. bovis* and *B. bigemina*, an economically important disease that affects cattle in tropical and sub-tropical countries [1,3]. Acute babesiosis in immunologically naïve cattle is characterized by high fever and severe anemia that can progress to either death or asymptomatic persistent infection [1]. The pathogenesis of acute *B. bovis* infection is also frequently accompanied by accumulation of parasite-infected RBCs in the microvasculature of vital organs, including the brain, resulting in cerebral babesiosis that resembles cerebral malaria in humans [18,19,20]. Live, blood- or in vitro culture-based, attenuated vaccines against *B. bovis* and *B. bigemina* are available in some endemic areas; however, these interventions have several constraints that prevent their widespread use [21,22,23,24,25]. Alternatively, anti-babesial drugs can be used to control acute bovine babesiosis, but this strategy is expensive and unpractical for large herds of cattle, in addition to raising concerns for potential development of drug-resistant parasites and antibiotic residues [26,27,28,29,30].

Development of efficacious and sustainable vaccines against human and bovine babesiosis is urgently needed; however, critical knowledge gaps remain on the protective immune responses of the vertebrate host to *Babesia* parasites. Importantly, concerning bovine babesiosis, young cattle (<1-year old) are resistant to *B. bovis* and *B. bigemina* acute disease and may become chronically asymptomatic reservoirs for parasite transmission [31]. This resistance of calves to acute babesiosis is associated with early activation of pro-inflammatory innate immune responses and is independent of passive immune maternal factors [32,33,34,35,36,37]. Therefore, myeloid innate immune cells, such as monocytes, macrophages, and dendritic cells (DC), may play a critical role in protection against *Babesia*. Collectively, results suggest that a balanced pro-inflammatory innate immune response, combined with production of nitric oxide (NO), can control parasitemia during acute infection, allowing the host to develop a protective acquired immune response [36,38,39,40,41,42,43]. However, definitive correlates of protection against human and bovine babesiosis remain largely unknown and represent an important knowledge gap for vaccine development.

Trained immunity (TI) is defined as a reprograming of the vertebrate innate immune system evoked by certain exogenous or endogenous insults, leading to immunological memory and more efficient responses to subsequent specific or non-specific challenges [44]. Interestingly, TI involves epigenetic and metabolic modifications of the vertebrate innate immune myeloid cells, specifically monocytes, macrophages, DC, and neutrophils [44,45,46]. Furthermore, results show that TI is independent of T and B cell responses [46]. Predictably, this emerging field of immunology has implications for the development of vaccines and therapeutics against infectious diseases. Of particular interest is the observation that some live attenuated vaccines induce TI, making innate immune cells more efficient in responding to heterologous challenges [44,46,47,48,49]. Therefore, this heterologous immunity caused by live vaccines can be explored to develop novel prophylactic and therapeutic control strategies against pathogens. *Mycobacterium bovis* bacillus Calmette-Guerin (BCG) is a live attenuated vaccine that has been used worldwide for more than 100 years to prevent tuberculosis (TB) in humans. With a long history of use against TB and excellent safety record, BCG is currently part of the World Health Organization immunization program [50]. A large body of recent evidence indicates that BCG induces TI that is characterized by the development of circulating monocytes, macrophages, and natural killer (NK) cells with increased capacity to produce pro-inflammatory cytokines [46,48,51,52]. Initial studies demonstrated that BCG TI induces cross-protection against *Candida albicans* and *Staphylococcus aureus*, among other unrelated pathogens [46,53]. Subsequent observational studies in children in West Africa demonstrated that BCG vaccination decreases morbidity due to infections other than TB, including malaria [54,55]. Very recently, epidemiological studies indicated that TI induced by BCG could be protective against the development of severe coronavirus disease 2019 in humans [56,57]. Strikingly, studies performed in the 1970’s showed that BCG inoculation protects mice against lethal experimental infection with *B. microti* and *Plasmodium berghei* [58,59]. At the time, no immunological mechanism was identified as responsible for this protective effect of BCG on *B. microti* and *P. berghei*, and no further investigations were reported [60]. Regardless, these results suggest that BCG TI may induce cross-protection against apicomplexan parasites, including *Babesia* and *Plasmodium* species.

Here in this review, we consider the re-emerging importance of BCG within the novel field of TI in conjunction with the evidence that this attenuated bacillus induces heterologous cross-protection, specifically against apicomplexan parasites. Our goal is to raise awareness of BCG TI in the context of *Babesia* infection and discuss the prospective application of BCG to train immune cells to control human and bovine babesiosis. We also discuss the possibility of using TI in conjunction with recombinant BCG (rBCG) vaccines expressing *Babesia* immunogens to generate acquired immunity as well.

## 2. Human and Bovine Babesiosis

Human babesiosis is an acute and persistent disease caused primarily by *B. microti* and *B. divergens*; however, emerging cases of infections with *B. duncani* and *B. divergens*-like parasites have also been reported in the USA [7,9,11,16,61,62]. Humans are accidental hosts of *B. microti* and most cases occur from late spring to early autumn when humans are in closer contact with tick vectors and definitive mammalian hosts, such as rodents and deer [16,63]. Worrisomely, *B. microti*, and potentially other *Babesia* species, can be transmitted to humans via blood transfusion, making it an important emerging threat to the blood supply worldwide [9,64,65]. No vaccine is currently available to protect humans against babesiosis, and the recent emergence in the USA of *B. microti* infections that are resistant to available therapeutics has raised serious concerns about the availability of treatments [66,67]. In addition, splenectomized and/or immunosuppressed individuals are more susceptible to *B. microti* infection that can progress into a life-threatening condition [11]. *B. microti* is a *sensu lato* species which is transstadially transmitted by tick vectors, and this biological characteristic needs to be considered for the development of efficient strategies to control the parasite. Altogether, these aspects demonstrate that a comprehensive assessment of the impact of human babesiosis on public health and the development of measures to control the disease are urgently needed.

Bovine babesiosis imposes draconian economic losses on cattle production in tropical and subtropical countries and represents a serious threat to food security worldwide [3,25,68,69]. Notably, *B. bovis* and *B. bigemina* are the most prevalent etiological agents of bovine babesiosis; however, the disease in bovids can also be caused by *B. divergens*, *B. occultans*, *B. major*, *B. orientalis*, and *B. ovata*, among other species [3,65,66]. Live-attenuated *B. bovis* and *B. bigemina* vaccines have been successfully used to control the devastating effects of acute disease in some endemic areas. Regardless of their relative efficacy, the *B. bovis* and *B. bigemina* attenuated live vaccines have several constraints that preclude their widespread use, especially for eradication programs in endemic areas and control strategies in non-endemic at-risk regions. Among the limitations of the live vaccines, there are the reliance on cattle for vaccine production, risk of co-transmission of unrelated blood pathogens, potential variation among vaccine batches, possibility of reversion to virulence, need for a cold chain for vaccine distribution, and establishment of persistent infection in vaccinated animals that can serve as reservoirs for parasite transmission [1,21,23,24]. These restrictions clearly demonstrate that improved, sustainable, and more efficient vaccines are needed to control bovine babesiosis.

Development of anti-babesial vaccines has been hampered by knowledge gaps in the identification of protective parasite antigens and the absence of definitive correlates of protection against the disease in vertebrate hosts. *Babesia* genome sequencing, combined with the development of transfection and gene editing methods, have advanced the knowledge of the parasites’ biology and its lifecycle. Despite the progress, critical gaps remain in our understanding of the parasite-host-tick interactions, which in part explains the current absence of effective subunit vaccines against babesiosis [70,71,72,73,74,75,76,77,78]. Furthermore, definitive correlates of protection against *Babesia* infection are unavailable and represent a major impediment for the development of efficient control strategies. Thus, a better understanding of parasite biology and the immunological mechanisms, especially the innate immune responses associated with protection, is a critical step toward the development of effective anti-babesial vaccines.

## 3. Immune Responses to *Babesia* spp.

Most of our current knowledge of immune responses to *Babesia* parasites has emerged from research in *B. bovis* and *B. microti* using cattle and mouse models, respectively. Collectively, these studies indicate that early activation of innate immunity is a key component for the vertebrate host to control parasitemia and survive acute infection [34,42,43,79,80,81]. Expression of pro-inflammatory cytokines, such as IL-12 and IFNγ, early during acute infection and consequent delay in the production of the regulatory cytokine IL-10, is crucial to control parasitemia and develop protective acquired responses. Pro-inflammatory cytokines can act in an autocrine or paracrine manner to activate monocytes and macrophages to produce NO, which has been shown to be babesicidal [38,40,43,82]. It is, therefore, predicted that control of parasitemia during the early pre-clinical stage of the infection is mediated by myeloid innate immune cells [39,42,83]. It has also been shown that natural killer (NK) cells are a potential source of IFNγ early during acute *Babesia* infection, which may play a role in fine-tuning the profile of cytokine expression and boost activation of monocytes, macrophages, and DC [34,36,37,80]. 

Despite the importance of innate immunity, development of acquired immune responses, such as activation of CD4^+^ T lymphocytes and B cells in the spleen, are necessary for the resolution of acute clinical disease and transition into chronic asymptomatic infection. Activation of acquired immune responses during *Babesia* infection leads to the production of IFNγ, mainly by CD4^+^ T lymphocytes, which activates additional myeloid and lymphoid immune cells [84,85]. Collectively, this process helps the resolution of acute infection during late clinical stages of disease and promotes the development of protective and memory responses [83,86]. The role played by humoral immunity in limiting acute bovine babesiosis remains controversial, considering that some young cattle resolve acute infection in the absence of antibodies. However, it is possible that humoral immune responses, combined with cellular acquired responses, may play a role in controlling parasitemia during the chronic asymptomatic phase of the infection [72,87,88].

Historically, it has been demonstrated that young cattle (<1-year old) survive acute disease caused by virulent *B. bovis* and *B. bigemina* strains, and may develop persistent infection, becoming resistant to re-infections during adulthood and reservoirs for parasite transmission [31]. In contrast, susceptible adult bovines (>1-year old) succumb to acute disease approximately 15 to 20 days after infection [32,33]. This scenario presents a unique opportunity to study immune responses in resistant calves and compare them with those in adult susceptible animals. Importantly, it is reasonable to consider that infection of young cattle and development of persistent asymptomatic disease favor both the parasite and vertebrate host. This situation is particularly observed for bovine babesiosis in endemically stable regions in tropical and subtropical areas worldwide [89,90,91,92]. Interestingly, splenectomy abrogates the resistance of young cattle to *Babesia* infection and aggravates the condition of adult animals, indicating that the spleen plays a crucial role in protection [93,94,95]. In fact, a specific splenic CD13^+^ DC population was identified in bovine as a potential source of IL-12 after exposure to *B. bovis* [36].

In a nutshell, the available data support the premise that protection against acute babesiosis is associated with early activation of pro-inflammatory innate immune responses. As the vertebrate host survives the acute phase of the disease, it is likely that acquired immune responses, primarily driven by CD4^+^ T cells, are essential for establishing persistent infection in clinically healthy individuals. Despite the progress in our understanding of *Babesia* immunology, specific immunological biomarkers of protection remain largely unknown. Considering that innate immunity is essential for controlling parasitemia during acute infection and survival of the vertebrate host, efficient activation of myeloid innate immune cells, perhaps via TI, emerges as a reasonable and possible approach to control acute babesiosis.

## 4. *Mycobacterium bovis* BCG 

BCG is a strain of *Mycobacterium bovis* that was empirically attenuated through several in vitro passages between 1908 and 1921 at the Pasteur Institute of Lille, France [96]. Subsequent evaluation of BCG in animal models demonstrated its infectivity yet significant attenuation. In the late 1920’s, BCG was recommended by the League of Nations as the official vaccine against human TB, and since then, it remains the only commercially available vaccine against the disease. BCG is currently the world’s most widely used vaccine and has been administered safely to more than three billion people. BCG offers unique advantages as a live attenuated vaccine: it is unaffected by maternal antibodies and therefore, it can be given at any time after birth; BCG is stable and safe; it is usually given as a single dose eliciting a long-lasting immunity; it can be administrated parenterally or orally; and it is inexpensive to produce when compared to other live vaccines. In addition, the strong adjuvanticity of BCG makes it an attractive vector for the development of recombinant vaccines against infectious diseases of vertebrates [97,98,99,100,101,102,103]. Despite the advantages and strong record of safety, the efficacy of BCG as an anti-TB vaccine remains controversial [104,105,106,107].

Recent analysis of the BCG genome demonstrated several deletions and rearrangements that likely happened during its continuous in vitro cultivation that led to attenuation [108,109,110,111]. One such deletion involved the loss of the region of difference 1 (RD1), which encodes for the 6-KDa early secretory antigenic target (ESAT-6) and the culture filtrate protein-10 kDa (CFP-10), among other genes [112,113]. As a result of the absence of RD1, after phagocytosis, BCG remains in the phagosome, in contrast to virulent *M. bovis*, *Mycobacterium leprae* and *Mycobacterium tuberculosis* that escape into the host cell cytosol [114]. This characteristic has implications for the establishment of persistent infection by BCG after vaccination, considering that some bacilli are killed inside the cell phagosomes [115]. Nevertheless, one unquestionable trait of BCG vaccination is the strong activation of pro-inflammatory innate immune responses, which leads to the development of a T helper type 1 (TH1)-like acquired immune response [116,117,118,119]. Moreover, once BCG was developed at the Pasteur Institute, visiting scientists took stocks to their respective countries of origin at various times, giving rise to several daughter strains or substrains, such as Birkhang, China, Connaught, Copenhagen, Denmark, Frappier, Glaxo, Japan, Mareau, Mexico, Pasteur, Phipps, Prague, Russia, Sweden, and Tice [120]. This led to different passage histories and the inevitable in vitro evolution of diverse strain genotypes and phenotypes. For instance, the current BCG Pasteur differs from the original Pasteur strain by polymorphisms and deletions. BCG strains are used both as a vaccine to prevent TB and as an immunotherapy of bladder cancer [57,121]. In addition, there is active research on the use of BCG to treat other cancers and non-infectious diseases, such as autoimmune disorders and type I diabetes [122]. Several studies have addressed how genetic differences among BCG substrains may be reflected in alterations of immunogenicity and protection against TB and non-mycobacterial diseases. However, reaching definitive conclusions has not yet been possible, due to confounding effects. Thus, although the substrain selection is an important consideration in any potential use of BCG TI, definitive guidelines have not been established. In the USA, two formulations of the strain Tice (developed by the University of Illinois from a strain originated at the Pasteur Institute) are approved against TB: BCG Vaccine U.S.P. for percutaneous use and BCG Live (TICE^R^ BCG) for intravesical use as described in the corresponding package inserts.

Following inoculation, BCG pathogen-associated molecular patterns (PAMPs), such as peptidoglycan, arabinogalactan, and mycolic acids located at the bacterial cell wall, are recognized by pattern recognition receptors (PRRs) of monocytes, macrophages, and DC. Among the surface PRRs that recognize BCG PAMPs are toll-like receptors (TLR) 2, TLR4, and mannose receptors [123]. In addition, complement receptor (CR) 3 and CR4 opsonize BCG, favoring recognition and phagocytosis of the bacillus by the innate immune cells. After phagocytosis, BCG PAMPs are also recognized by cytosol receptors, such as nucleotide-binding and oligomerization domain (NOD)-like receptors (NLRs), and C-type lectins, retinoic acid-inducible gene I (RIG-I)-like receptors. Combined, these events lead to activation of myeloid cells, which is characterized by up-regulation of co-stimulatory surface molecules CD40, CD80, and CD86 [124,125,126]. Also, BCG-activated innate immune cells up regulate the expression of pro-inflammatory cytokines, mainly IL-12, IL-1β, TNFα, IL-6, and IL-8, and migrate to regional lymphoid organs where they act as antigen-presenting cells to lymphocytes, initiating the acquired immune responses [125,127]. As a result, CD4^+^ and CD8^+^ T lymphocytes produce large amounts of IFNγ, which in turn, induces further activation of myeloid cells to up regulate the expression of pro-inflammatory cytokines.

In addition to eliciting pro-inflammatory immune responses, BCG also reprograms the host metabolic pathways, which has a profound impact on how individuals respond to related and non-related pathogens. It has been demonstrated that BCG switches the cell metabolism from oxidative phosphorylation to glycolysis, therefore affecting the development of protective immunity [128]. The combined effects of BCG vaccination on immune responses and metabolism may mechanistically explain the heterologous protection induced by the bacillus and its TI characteristic.

## 5. BCG and Trained Immunity

Recent studies have demonstrated that certain live attenuated vaccines induce TI, defined as the development of immunological memory in classical innate immune cells [44,129,130,131]. However, TI differs from innate immunity as it is expressed in response to a secondary infection following primary infection by an unrelated pathogen or vaccination with live-attenuated vaccines. This paradigm shift sheds light on potential mechanisms for the observed effects that live vaccines have on unrelated pathogens. Of particular interest is the fact that BCG induces TI; however, this observation has also been reported after inoculation with other live vaccines, such as those against measles, smallpox, and the Sabin polio vaccine [47,132]. An important body of work indicates that BCG induces TI in myeloid innate immune cells by epigenetic mechanisms involving DNA methylation, histone modification, and expression of non-coding RNA [53,132]. In addition, BCG induces metabolic changes in immune and non-immune cells of vaccinated hosts, which may have an impact on how the cells respond to pathogens [44,51,129,130,133]. Initial studies showed that BCG TI was responsible for heterologous protection against *Candida albicans* and *Staphylococcus aureus* [46]. The beneficial non-specific effects of BCG TI were also demonstrated by a significant reduction of newborn deaths due to mycobacterial unrelated infectious diseases in BCG vaccinated populations in West Africa [54].

After BCG vaccination, innate immune cells are activated by bacterial cell wall components, and this cell-pathogen recognition mechanism is behind the well-described ability of the bacillus to serve as an immune adjuvant [124,125]. BCG-activated monocytes, macrophages, and DC produce pro-inflammatory cytokines, which can have an effect against unrelated pathogens [48,51]. In addition, cells activated by BCG develop a TI profile, which in monocytes is characterized by histone modifications with methylation of lysine at positions 4 and 9 in the histone 3, H3K4me3 and H3K9me3, respectively, within the promoters of the TNFα, IL-6, TLR4, and IL-1β genes [128]. These alterations lead to up regulation of these specific genes and development of an efficient pro-inflammatory response upon subsequent encounter with related or unrelated pathogens [46,51]. Interestingly, it has been shown that BCG vaccination can also induce epigenetic changes in bone marrow progenitor myeloid cells, generating myelopoiesis and trained cell populations highly equipped to respond to a variety of pathogens [134,135]. This mechanistically explains how BCG TI can potentially have a long-lasting effect regardless the short life of monocytes, macrophages, DC, and neutrophils. In fact, BCG TI has been shown to have a long-term effect on innate immune cells that ultimately drives efficient TH1 and T helper 17 (TH17) acquired responses [46,136].

BCG vaccination was shown to provide protection against bovine tuberculosis in cattle [137]. However, the purified protein derivative (PPD) skin assay is considered the gold standard diagnostic test for TB in vertebrates, including humans and livestock. Unfortunately, BCG vaccination usually induces positive PPD results. Several studies have been performed to overcome this limitation. Some approaches focused on the use of DIVA (differentiating infected from vaccinated animals) diagnosis. Tests with significant sensitivity and specificity have been developed based on antigens or peptide mixtures that are not present (ESAT-6, CFP10) or not secreted by BCG (Rv3615c) that can be used both as IFNγ blood or skin tests [138]. Moreover, the utility of these tests was evaluated in field studies with promising results in low- and middle-income countries [139,140]. Other investigations attempted the development of BCG strains that are unable to induce PPD positive results. One study showed that a BCG *leuD* auxotrophic mutant was able to protect guinea pigs without eliciting a PPD response [141]. However, the corresponding *M. bovis leuD* mutant, while providing protection in cattle, still elicited a strong PPD response [142]. Another approach combined both DIVA antigens with the development of a BCG triple deletion mutant that ablated five BCG genes (3043, 2895, 2897, 3679, and 3680). The antigens encoded by the deleted genes were then used in cocktails to demonstrate their DIVA potential in guinea pigs [143]. Although the mutants described herein were marked with drug-resistant determinants, several technologies have been developed to generate multiple unmarked mutants [144,145]. The challenge remains in optimizing protection while maintaining DIVA capabilities in cattle. Nonetheless, these developments open several opportunities for the widespread use of BCG TI to induce heterologous protection against pathogens, without interfering with TB control programs.

Considering the resurgence of BCG as an TI inducer and the crucial role that innate immunity plays in initiating and driving acquired protective immune responses, the impact of BCG TI needs to be further investigated in the context of relevant infectious diseases of vertebrates. Therefore, the use of BCG TI to help protect against *B. microti* and *B. bovis* infections, and other apicomplexans, emerges as a rationale and doable option, especially considering the current absence of efficacious and sustainable approaches to control human and bovine babesiosis.

## 6. BCG Trained Immunity in the Context of *Babesia* Infection

Considering the ability of BCG to train innate immune cells and the importance that the innate immune system has in protecting vertebrates against *Babesia* infection, we revisited seminal studies performed in the 1970’s which demonstrated that BCG inoculation protected mice against lethal challenge with *B. microti* and *P. berghei*. [58,59]. Interestingly, these studies demonstrated that the BCG-mediated protection against *B. microti* resulted in intracellular death of the parasite, which was not mediated by antibodies or increased phagocytic activity of monocytes/macrophages. At the time, no mechanistic explanation was available for the observed protection of BCG against *B. microti*. Here we propose a possible model to reinterpret those early observations considering the novel discoveries indicating that BCG TI induces epigenetic and metabolic changes that may control infections with apicomplexan parasites.

BCG vaccination induces a massive and rapid elicitation of the innate immune responses characterized by activation of monocytes, macrophages, DC, and neutrophils [118,124,125,126]. Studies in humans and mice have also shown that BCG inoculation induces NK cells to secrete IFNγ, which ultimately enhances the activation of myeloid innate immune cell populations [146,147]. BCG activation of the innate immune cells drives primarily the development of a pro-inflammatory TH1 acquired immune response, with marked production of IFNγ primarily by CD4^+^ T cells and clonal expansion of B cells [116,117,119]. Activation of innate immunity and development of a TH1 profile induced by BCG vaccination markedly modulates the entire immune responses of vertebrates and can affect both related and unrelated pathogens. Regarding apicomplexan parasite infections, it was shown that BCG inoculation can prevent experimental cerebral malaria in mice [148]. In addition, BCG vaccination modulates clinical, immunological, and parasitological features, favoring the control of malaria infection in humans [55]. One study demonstrated that BCG protected mice against *P. yoelii* infection by shifting the immune response toward TH1 type and induction of protective IgG2a [149]. Collectively, it is likely that the development of TI following BCG vaccination is potentially the mechanism behind the unspecific effect of this attenuated bacillus against unrelated pathogens, including apicomplexan parasites. This hypothesis needs to be further investigated considering the recent knowledge of BCG TI and its potential implication in controlling human and bovine babesiosis. Therefore, we postulate that epigenetic reprograming caused by BCG would allow myeloid cells to respond faster and more efficiently to *Babesia* infection, which in turn, could protect against the acute phase of the disease. Once activated, BCG TI mechanisms could help the innate immunity control parasitemia during acute infection, buying time for the vertebrate host to develop a protective acquired immune response. This hypothesis is supported by the studies of *B. microti* and *P. berghei* in the mouse model [58,59]. The premise is also reinforced by previous observations in the cattle-*B. bovis* model indicating that early activation of innate immune responses, with consequent production of pro-inflammatory cytokines, is essential for the host to survive acute infection [34,42,43,83]. Therefore, the innate immune responses elicited by BCG vaccination and subsequent elicitation of TI are the same ones required for protection against *Babesia* acute infection. In Figure 1 we propose a model of BCG TI and its potential implication on *Babesia* infection. Table 1 summarizes the BCG TI and its potential effects on *Babesia*.

A recent study demonstrated that BCG aerosol vaccination induces the development of a TI phenotype in bovine monocytes [52]. Results from this study showed a significant increase in the expression of pro-inflammatory cytokines and metabolic changes in monocytes from BCG-exposed cattle compared to cells from control animals. In addition, data indicate that BCG TI induced functional changes in bovine monocytes, which were characterized by increased transcription of pro-inflammatory cytokines upon stimulation with TLR agonists [52]. Collectively, the *B. bovis*-cattle model may be uniquely suited to investigate the usefulness of BCG TI against relevant cattle diseases, such as bovine babesiosis. In this context, it remains to be determined if BCG TI can control *Babesia* parasitemia during acute infection in cattle, allowing the host to mount protective acquired immune responses. Also, studies are needed to investigate phenotypic and epigenetic differences in myeloid cells between BCG-vaccinated and naïve individuals and the implications for susceptibility and resistance to acute babesiosis. These investigations may reveal novel innate immune correlates of protection that can be exploited to design effective anti-*babesial* vaccines.

## 7. Recombinant BCG (rBCG) and the Potential to Induce TI and Adaptive Immunity

There is a long history of modifying BCG to serve as a vectored vaccine to carry protective antigens from other pathogens. Pioneering studies demonstrated expression of β-galactosidase, tetanus toxin fragment C, HIV Gp120, HIV Gag, and HIV Env proteins using replicating and/or integrating vectors carrying a heat shock promoter [155,156]. These rBCG strains elicited both humoral and cell-mediated immunity. Further developments included the construction of appropriate vectors to express antigens either in the cytoplasm, cell surface or as fusion lipoproteins in BCG that have been applied to both human and veterinary viral, bacterial, and parasitic pathogens [98,101,157,158,159]. rBCG strains have also been developed as immunotherapeutic expressing cytokines, tumor antigens or other non-mycobacterial antigens to treat bladder cancer and other tumors [122]. Auxotrophic mutants that work in concert with auxotrophic markers were also developed to avoid drug-resistance and provide stable maintenance of both integrating and multi-copy plasmids in vivo [160,161,162,163]. In addition, reporter genes are available to monitor gene expression in vivo and regulated promoter systems that can be fine-tuned in vivo [164,165,166]. This toolbox provides us with ample opportunities to construct rBCG strains that could first induce TI against *Babesia* at early-times post infection followed by the induction of specific protective antigens that could induce acquired immunity at later times following the parasite infection. In fact, the ability of BCG to deliver the *B. bovis* rhoptry associated antigen 1 (RAP-1) was successfully evaluated in mice, supporting the hypothesis that rBCG can be employed as a component of anti-babesial vaccines [167]. Considering that the referred study was performed in mice, it remains to be determined if rBCG expressing babesial antigens can protect cattle against babesiosis.

## 8. Conclusions

In a seminal paper published in 1933 [96], Dr. Albert Calmette asked, “Does the harboring of BCG confer on the organism a special aptitude to resist those other infections which are so frequent in young children?” Dr. Calmette’s question was raised in relation to the potential effects of BCG vaccination on protecting infants from non-tuberculosis infections. Today, we can certainly answer the question in an affirmative way considering what we now understand of the TI mechanisms induced by BCG vaccination. In this regard, it becomes clear that understanding the specific immune pathways and epigenetic signatures of BCG TI is essential to develop novel therapeutic control strategies against relevant infectious diseases of vertebrates. We predict that BCG TI and rBCG could be new and efficient approaches to induce protection against acute babesiosis in humans and cattle. This strategy can be used as a solo method to control acute infection and prevent deaths, either in combination with specific anti-babesial vaccines or using a specific rBCG as indicated above, to also prevent the establishment of persistent infection. This single or dual strategy could potentially decrease parasite load in the vertebrate reservoir host, diminishing the potential for parasite transmission and help control the disease. Additionally, by better preparing the vertebrate innate immunity to control babesiosis, BCG TI can also play an auxiliary role in programs to eradicate the disease. For instance, BCG can be used in conjunction with anti-babesial and/or anti-tick vaccines, or rBCG carrying babesial antigens, to prevent the development of blood and sexual stages of the parasite. The success of this strategy will simultaneously need the development of approaches to decrease and/or eliminate the parasite in tick vectors, such as transmission-blocking vaccines. In conclusion, we propose the use of the *B. microti*-mouse and *B. bovis*-cattle models to address pertinent questions regarding the effect of BCG TI and rBCG on human and bovine babesiosis, respectively. For instance, if BCG TI is efficient in controlling acute *Babesia* infection, what is the immunological mechanism(s) involved in protection? What is the duration of the effect? Do individuals vaccinated with BCG and subsequently infected with *Babesia* become persistently infected with the parasite? If so, can they serve as reservoirs for parasite tick acquisition and transmission? Is rBCG expressing a specific set of cytokines more efficient than wild type BCG in training the vertebrate innate responses against *Babesia*? What is the BCG-mediated epigenetic landscape associated with protection against acute babesiosis? Can different routes of BCG delivery enhance protection against *Babesia* parasites? Can BCG TI help overcome the difficulties of achieving heterologous protection with anti-babesial subunit and attenuated vaccines? Addressing these questions about BCG TI in the context of *Babesia* infection will certainly open new opportunities for the development of sustainable and efficient strategies to control human and bovine babesiosis and provide crucial leads to control other important human and veterinary diseases.

## Figures and Tables

**Figure 1 vaccines-10-00123-f001:**
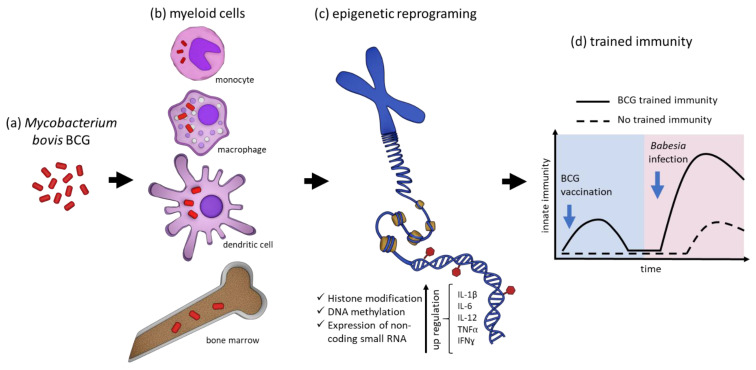
Trained immunity (TI) induced by *Mycobacterium bovis* BCG and its implication on *Babesia* infection. (**a**) Upon inoculation, BCG is phagocytosed by (**b**) myeloid cells, primarily monocytes, macrophages, and dendritic cells (DC), and myeloid precursor cells in the bone marrow. (**c**) BCG can survive inside the cells and induce TI by epigenetic reprograming mechanisms, which are characterized by histone modifications, DNA methylation in target genes, and expression of non-coding small RNA. (**d**) We hypothesize that epigenetic reprograming would allow these innate immune cells to respond faster and more efficiently to *Babesia* infection, which in turn, will elicit protection against acute babesiosis.

**Table 1 vaccines-10-00123-t001:** BCG trained immunity and its potential effects on *Babesia* infection.

BCG Trained Immunity	Potential Effect on *Babesia* Infection
Activation of myeloid innate immune cells to up regulate pro-inflammatory cytokines and inflammasome pathways [44,46,51,130,131].	Autocrine and paracrine activation of innate immune cells by pro-inflammatory cytokines that can control parasitemia early during *Babesia* acute infection [39,40,42,43,83].
Priming myeloid innate immune cells to produce reactive nitrogen species (RNS) [150,151].	Induction of babesicidal RNS, including NO [39,40,41].
BCG vaccination switches the metabolisms of immune and non-immune host cells from oxidative phosphorylation to glycolysis [133].	Metabolic alterations may affect the development of protective immunity against apicomplexan parasites [152]; for instance, more availability of arginine to innate immune cells and RBCs may be associated with NO production and control of *Babesia* parasitemia, as demonstrated in *Plasmodium* [153,154].

## Data Availability

Not applicable.

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
