# Peer review of "Harnessing Mycobacterium bovis BCG Trained Immunity to Control Human and Bovine Babesiosis"

_vaccines, 2022, doi:10.3390/vaccines10010123_

Round 1

Reviewer 1 Report

The main question addressed by the research is potential development of an enhanced vaccine against cattle and human babesiosis, using Mycobacterium bovis BCG, a long-time used as anti-TB vaccine, as vector for expressing babesia antigen, and the training immune properties of this mycobacteria in inducing innate immunity against other pathogens. 

It is highly relevant in the field, especially with babesiosis, for which there exists only one type of vaccine, a live and attenuated parasite (for animals only, not humans), which must be grown in vivo or more expensively, in vitro. The approach proposed by the authors is on a vectored bacteria already used as vaccine for now a long time hence, the production and delivery will be certainly easier than the present one. 

The manuscript calls the attention to a method of vaccination that might be more protective than others. 

This is a review on the potential use of the mycobacteria as vector for vaccine development and will certainly aid in the expansion of the field. 

The conclusions are consistent with the evidence and arguments presented. The references are appropriate.

If such vaccine is successful, and apparently has an enormous change of success, it will be applicable to other similar pathogens, like malaria parasites. The review comes at the right time. 

Author Response

Reviewer #1

The main question addressed by the research is potential development of an enhanced vaccine against cattle and human babesiosis, using Mycobacterium bovis BCG, a long-time used as anti-TB vaccine, as vector for expressing babesia antigen, and the training immune properties of this mycobacteria in inducing innate immunity against other pathogens. 

It is highly relevant in the field, especially with babesiosis, for which there exists only one type of vaccine, a live and attenuated parasite (for animals only, not humans), which must be grown in vivo or more expensively, in vitro. The approach proposed by the authors is on a vectored bacteria already used as vaccine for now a long time hence, the production and delivery will be certainly easier than the present one. 

The manuscript calls the attention to a method of vaccination that might be more protective than others. 

This is a review on the potential use of the mycobacteria as vector for vaccine development and will certainly aid in the expansion of the field. 

The conclusions are consistent with the evidence and arguments presented. The references are appropriate.

If such vaccine is successful, and apparently has an enormous change of success, it will be applicable to other similar pathogens, like malaria parasites. The review comes at the right time. 

Response:

We deeply appreciate the positive comments from the reviewer. We agree with the reviewer that the use of BCG, either the wild type or recombinant bacterium, has a great potential to elicit a balanced activation of the innate immune responses in the vertebrate host and consequently control the Babesia parasitemia and the onset of acute disease.   

Reviewer 2 Report

Cogent, well constructed, well supported by the available evidence, and very well written.  Trivial errors of punctuation, syntax, or spelling , together with a note of a recent memoir

Congratulations!

note:

  1. line 69:constraints  "constrains" is the verb
  2. line 96:“Of particular interest, it is ”, delete“it”
  3. line 104:indicate add s
  4. line 118:you may wish to consult: Clark IA.  Int J Parasitol. 2021 Oct 30:S0020-7519(21)00298-8. doi: 10.1016/j.ijpara.2021.10.002. Epub ahead of print. PMID: 34757090.
  5. line 241:er, parenterally
  6. line 301: change “ oxidative phosphorylation to glycolysis therefore, affecting the  development of protective immunity” into“from oxidative phosphorylation to glycolysis, therefore affecting the development of protective immunity ”
  7. line 360:“”Although, the mutants “ delete“,”
  8. line 405:change “BGC” into“BCG”
  9. line 409:“mechanisms could help the innate immunity control”,delete “the”

Author Response

Reviewer #2

Cogent, well constructed, well supported by the available evidence, and very well written.  Trivial errors of punctuation, syntax, or spelling , together with a note of a recent memoir

Congratulations!

Response:

It is very much appreciated the positive comment from the reviewer. We have made corrections in lines 114-117 to address the reviewer’s observation on the superb, recently published memoir of Dr. Ian Clark. Also, we have corrected the minor errors highlighted by the reviewer, as described below.

note:

1. line 69:constraints  "constrains" is the verb

Response:

The mistake was corrected.

2. line 96:“Of particular interest, it is ”, delete“it”

Response:

We deleted “it”.

3. line 104:indicate add s

Response:

We added “s” to “indicate.”

4. line 118:you may wish to consult: Clark IA.  Int J Parasitol. 2021 Oct 30:S0020-7519(21)00298-8. doi: 10.1016/j.ijpara.2021.10.002. Epub ahead of print. PMID: 34757090.

Response:

We appreciate the reviewer’s observation on the recently published memoir paper (Clark IA 2021). In fact, the mechanism(s) behind the protective effect of BCG on Babesia and Plasmodium remains an important knowledge gap that has been recently discussed considering the new concepts of trained immunity. Once revealed, this mechanism(s) could be explored to control babesiosis, malaria and other haemoprotozoan diseases.

Lines 114-117 were modified to address the reviewer’s comment as follows: At the time, no immunological mechanism was identified as responsible for this protective effect of BCG on B. microti and P. berghei, and no further investigations were reported (Clark, I.A. Int. J. Parasitol. 2021, 51, 1265-1276, doi:10.1016/j.ijpara.2021.10.002.).”

5. line 241:er, parenterally

Response:

The mistake was corrected.

6. line 301: change “ oxidative phosphorylation to glycolysis therefore, affecting the  development of protective immunity” into“from oxidative phosphorylation to glycolysis, therefore affecting the development of protective immunity ”

Response:

The mistake was corrected.

7. line 360:“”Although, the mutants “ delete“,”

Response:

We deleted the extra “comma.”

8. line 405:change “BGC” into“BCG”

Response:

We corrected “BGC” to “BCG”.

9. line 409:“mechanisms could help the innate immunity control”,delete “the”

Response:

The mistake was corrected.

Reviewer 3 Report

In this study, the authors are presenting data about the prospective application of Mycobacterium bovis bacillus Calmette-Guerin (BCG) to train immune cells to control human and bovine babesiosis.

The data discussed in this review are important and original to understand the potential use of trained immunity  in conjunction with recombinant BCG vaccines expressing Babesia immunogens, by opening new opportunities for the development of efficient strategies to control human and bovine babesiosis and providing crucial leads to control other important human and veterinary diseases.

The review is well supported by other published materiale and early investigations demonstrating that BCG inoculation protects mice against experimental B. microti infection and recent observations that BCG vaccination  decreases the severity of malaria in children infected with Plasmodium falciparum, a Babesia-related parasite.

The conclusions are well consistent with the evidence and arguments presented.

The references result appropriate.

The tables and figures are well elaborated.

Minor comments:

L 22: Please remove acronym (TI)

L 26: Please remove acronym (BCG)

L104: immunization program instead of Immunization Program

Author Response

Reviewer #3

In this study, the authors are presenting data about the prospective application of Mycobacterium bovis bacillus Calmette-Guerin (BCG) to train immune cells to control human and bovine babesiosis.

The data discussed in this review are important and original to understand the potential use of trained immunity  in conjunction with recombinant BCG vaccines expressing Babesia immunogens, by opening new opportunities for the development of efficient strategies to control human and bovine babesiosis and providing crucial leads to control other important human and veterinary diseases.

The review is well supported by other published materiale and early investigations demonstrating that BCG inoculation protects mice against experimental B. microti infection and recent observations that BCG vaccination  decreases the severity of malaria in children infected with Plasmodium falciparum, a Babesia-related parasite.

The conclusions are well consistent with the evidence and arguments presented.

The references result appropriate.

The tables and figures are well elaborated.

Response:

We sincerely thank the reviewer for the very positive comments.

Minor comments:

L 22: Please remove acronym (TI)

L 26: Please remove acronym (BCG)

Response:

We appreciate the reviewer’s comment; however, we would like to maintain the acronyms for “TI” and “BCG” in lines 22 and 26, respectively. Our reason for this request is based on the fact that both “trained immunity (TI)” and “bacillus Calmette-Guerin (BCG)” are repetitively mentioned in several lines of the Abstract (lines 26, 27, 28, 30, and 32). Therefore, it would make more sense to use the “acronym format”. We believe that the use of these acronyms help the flow the abstract and facilitate the understanding of the text by readers.

L104: immunization program instead of Immunization Program

Response:

The mistake was corrected.